# A Reliable Observation Point Selection Method for GB-SAR in Low-Coherence Areas

Zexi Zhang [ID], Zhenfang Li *, Zhiyong Suo [ID], Lin Qi [ID], Fanyi Tang [ID], Huancheng Guo [ID] and Haihong Tao [ID]

National Key Laboratory of Radar Signal Processing, Xidian University, Xi'an 710071, China; zexizhang@stu.xidian.edu.cn (Z.Z.); zysuo@xidian.edu.cn (Z.S.); 20021110078@stu.xidian.edu.cn (L.Q.); 18021210100@stu.xidian.edu.cn (F.T.); 21021110328@stu.xidian.edu.cn (H.G.); hhtao@xidian.edu.cn (H.T.)
* Correspondence: lzf@xidian.edu.cn; Tel.: +86-029-8820-1933

**Abstract:** Ground-Based Synthetic Aperture Radar (GB-SAR), due to its high precision, high resolution, and real-time capabilities, is widely used in the detection of slope deformations. However, the weak scattering coefficient in low-coherence areas poses a great challenge to the observation point selection accuracy. This paper introduces a selection process for reliable observation points that integrates phase and spatial information. First, for various observation points with differentiated stability, we propose to utilize maximum likelihood estimation (MLE) methods to achieve stability assessment. Second, a phase correction approach is proposed to address unwrapped phase errors encountered at less stable points. Third, adaptive filtering for deformation information at observation points is achieved using estimated variance combined with wavelet filtering thresholds. Finally, in dealing with unknown deformation trends, we propose utilizing a clustering method to accurately identify these trends, thereby enhancing the precision in identifying reliable observation points (ROPs). The experimental results demonstrate that this method enhances the accuracy of observation point selection in low-coherence areas, providing a broader observational field for deformation detection.

**Keywords:** GB-SAR; slope deformation detection; maximum likelihood estimation; low-coherence areas; DBSCAN clustering method

## 1. Introduction

The successful application of spaceborne synthetic aperture radar (SAR) technology [1–3] sparks interest in exploring higher precision and more detailed structural information of targets, promoting the development of Ground-Based Synthetic Aperture Radar (GB-SAR) technology [4–7]. GB-SAR, with its unique advantages over its counterparts, has carved a niche in specialized applications. It offers superior spatial resolution and boasts versatile deployment strategies. Although GB-SAR faces challenges in terms of spatial coverage, its exceptional precision, real-time processing capabilities, and localized specificity are invaluable in certain application contexts. Through continuous monitoring [8], GB-SAR provides more comprehensive structural information on targets, facilitating precise deformation measurements of local areas. Additionally, the GB-SAR system, based on multiple input multiple output (MIMO) technology [9,10], proves effective in detecting displacements in large structures such as dams [11], bridges [12,13], buildings [14,15], and inverting the velocity fields of landslides [16,17].

In deformation analysis [18], GB-SAR frequently faces challenges in low-coherence areas with bare soil or vegetation. The typical characteristics of those areas are low-coherence, weak scattering intensity, and high signal phase noise, which cause difficulties in observation point selection and deformation analysis [19]. In deformation measurement applications, "Permanent Scatterers (PS)" with relatively stable reflection properties are commonly chosen for observation and analysis. This method was first proposed [20] by Ferretti et al. to overcome the drawbacks brought about by time [21] and geometric decorrelation [22]. Despite its widespread adoption [23,24], the PS method, which relies on

amplitude discretization to pinpoint permanent scatterers, falls short in addressing phase instability issues, prevalent in areas of low coherence. Subsequently, improved methods for pixel selection based on sublook spectral correlation [25], the top eigenvalue of coherence matrix [26], and the maximum likelihood theory were developed [27]. Currently, these techniques are mainly applicable to spaceborne synthetic aperture radar [28,29] application scenarios with relatively long temporal baselines [30], large resolution cells [31], and high coherence [32]. In the GB-SAR fields, the accuracy of observation point identification remains limited. The GB-SAR monitoring approach [33] introduces systematic error and atmospheric error, but it lacks a comprehensive strategy for addressing the phase errors associated with low-coherence areas. Image interpolation [34] or interference filtering in signal processing leads to errors and fails to apply well to GB-SAR with low-coherence coefficients and point scattering.

GB-SAR demands greater stability from observation points, as phase instability can lead to unwrapped phase errors, thereby substantially impacting the accuracy of measurement outcomes. Filtering techniques utilized to mitigate the issue of high noise in low-coherence scenarios fall into two main categories: spatial filtering [35–37] and temporal filtering [38,39]. In spatial filtering scenarios characterized by relatively high noise levels, the high coherence point phase is susceptible to the influence of the surrounding random phase. Consequently, the spatial filtering method, as detailed in literature [35], is not adapted to higher noise application scenarios. In time-domain filtering, Gaussian noise and mutation phases tend to contain wider spectral distributions. It is well known that frequency domain filters [40] are usually designed for specific frequencies. However, effectively filtering Gaussian noise and mutation by frequency domain filters is challenging. The Kalman filtering [38] in time-domain filtering methods proves effective, but it is difficult to adaptively optimize observation points with different stability in the detection area. Therefore, a methodology is needed to solve the problems encountered in the low-coherence areas.

The deformation analysis in low-coherence areas faces two important challenges. The first one is to improve the density and phase stability of observation points, and the second one is to screen observation points with correct deformation trends. To address the first problem, this paper estimates the distribution parameters of the phase information based on maximum likelihood estimation (MLE), which can reflect the stability and reliability of the observation points. The preliminary screening of observation points needs to address the unwrapped phase error caused by the phase mutation information and filter the phase information based on the threshold value to improve the stability of the phase. It is worth noting that there is a differentiation in the stability of observation points in different areas. The filter parameters are adaptively adjusted according to the MLE results so that the filter achieves the optimal filtering state compatible with each observation point. For the second problem, since MLE cannot estimate the deformation trend of each observation point, further processing is needed to filter the correct deformation trend. These filtered observation points include both reliable targets such as rocks or buildings, and targets with unreliable deformation trends, such as trees or vegetation. The deformation trends of reliable points show similarity and consistency. Conversely, the deformation information of unreliable points shows random trends. Based on the above basic regularity, this paper realizes the selection of reliable observation points (ROPs) by combining the Density-Based Spatial Clustering of Applications with Noise (DBSCAN) algorithm and the Lyddane–Shindo criterion. In addition, atmospheric phase information can be estimated by fitting these ROPs with a minimum mean square error (MMSE) criterion [41].

The organization of this paper is as follows: Section 2 describes the working principle of MIMO GB-SAR and analyzes the phase information. Furthermore, the paper provides a detailed explanation of both MLE and wavelet filters. Section 3 describes the entire signal processing flow for deformation detection, discussing the application of DBSCAN clustering and the Lyddane–Shindo criterion in this processing method. Section 4 presents an analysis and discussion of the experimental results, and Section 5 concludes the document.

## 2. Phase Information Estimation and Filtering

### 2.1. MIMO GB-SAR and Phase Analysis

GB-SAR is an advanced technology for capturing target scattering information by transmitting and receiving electromagnetic waves. Generally, GB-SAR has one transmitter unit and one receiver unit and is fixed on a mechanical rail. The GB-SAR accomplishes SAR imaging by moving the rail, as shown in Figure 1a. SAR enhances imaging resolution by synthesizing a large "virtual" aperture through the collection of reflected signals from the same area at different positions. Similar to SAR imaging, GB-SAR transmits and receives echo signals at different locations on the rail to realize aperture synthesis.

The GB-SAR system used in this paper is based on MIMO technology and has multiple transmitter units and multiple receiver units. One transmitting unit and one receiving unit can be equivalent to a virtual center. In far-field conditions, the transceiver units work at different positions, which is equivalent to working at the virtual center. The reasonable design of the transmitter unit and receiver unit position can realize the virtual center in a straight line and equal interval distribution with the schematic diagram shown in Figure 1b. Similar to the rail GB-SAR, the MIMO GB-SAR sequence works along the virtual center position to complete the imaging. The MIMO GB-SAR has a higher image acquisition rate than the rail GB-SAR (collectively referred to as GB-SAR).

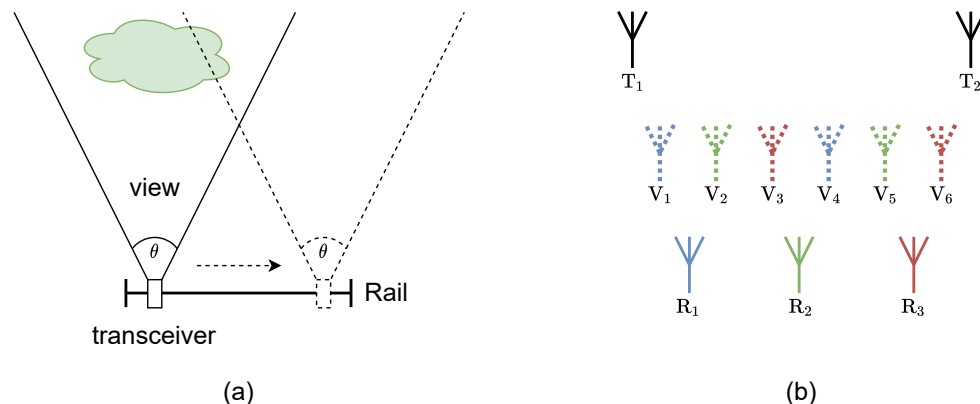

(a) (b)

**Figure 1.** (**a**) GB-SAR working diagram. (**b**) MIMO equivalent diagram. Transmit array (**top**), receiving array (**bottom**), and the virtual center array (**middle**).

GB-SAR operates at a fixed position, so the temporal baseline and observation scenarios are constant. When collecting data, image information of the same area is acquired over a continuous time sequence, denoted as $t_1, t_2, \ldots, t_N$. This data acquisition method allows us to capture the changes and movements of the target at different times. The information obtained from the acquired images can be depicted as a set of images denoted as $IM$, which can be expressed as

$$IM = [IM_1, IM_2, \ldots, IM_N]. \tag{1}$$

The phase of one point $p$ in the $IM$ can be expressed as

$$\phi_p(t) = \phi_{dis}(t) + \phi_{atm}(t) + \phi_{noise}. \tag{2}$$

In this model, the phase information is divided into three key components: $\phi_{dis}(t)$, $\phi_{atm}(t)$, and $\phi_{noise}$. The first part, $\phi_{dis}(t)$, represents the surface objects' deformation information caused by various geological and anthropogenic factors. The second part, $\phi_{atm}(t)$, is the atmospheric phase information, which cannot be neglected in remote sensing data because the refraction effect of the atmosphere can lead to waveform fluctuations. The third part, $\phi_{noise}$, denotes phase noise information, mainly caused by sensor noise, electro-

magnetic interference, and other factors, which may cover up the signals of fundamental ground changes. The feature deformation information $\phi_{dis}(t)$ can be represented as follows

$$\phi_{dis}(t) = \phi_{trend}(t) + \phi_{period}(t) + \phi_{mutation}(t), \tag{3}$$

where $\phi_{trend}(t)$ is the deformation trend, representing the ground objects' persistent trend over time, reflecting long-term internal evolution or external gradual change. $\phi_{period}(t)$ represents periodic deformation information, which refers to occasional fluctuations caused by seasonal changes in ground objects over a long period [42]. $\phi_{mutation}(t)$ denotes mutation information that characterizes sudden changes in the feature over a short period, which usually cause phase unwrapped errors.

The atmospheric phase can be expressed as

$$\phi_{atm}(t) = \frac{4\pi}{\lambda} \int_0^{r_p} \eta(t)dr, \tag{4}$$

where $r_p$ denotes the electromagnetic wave propagation path between the target and the radar. $\eta(t)$ is the atmospheric refractive index, which, according to the latest definition of the International Telecommunication Union [43], can be expressed as

$$\eta(t) = \left( 77.6\frac{P_d}{T_a} + 72\frac{e}{T_a} + 3.75 \times 10^5 \frac{e}{T_a^2} \right) \times 10^{-6} + 1. \tag{5}$$

Water vapor pressure $e$ is expressed as

$$e = \frac{H}{100}e_{sat}, \tag{6}$$

$$e_{sat} = 6.1121 \cdot exp\left[ \frac{\left\{18.678 - \frac{t}{234.5}\right\} \cdot t}{t + 257.14} \right] \cdot \left\{ 1 + 10^{-4}\left[ 7.2 + P \cdot \left( 0.0320 + 5.9 \cdot 10^{-6} \cdot t^2 \right) \right] \right\}, \tag{7}$$

where $P_d$ represents dry atmospheric pressure (hPa), and the unit of water vapor pressure $e$ is $hPa$. $T_a$ is the absolute temperature (K). $e_{sat}$ is the saturation vapor pressure (hPa). $H$ is the relative humidity (%). $t$ is the temperature (°C). $P$ is the total atmospheric pressure (hPa). Atmospheric pressure, humidity, and temperature vary with time, resulting in non-uniform electromagnetic characteristics. This will cause electromagnetic waves' speed and propagation direction to change as they travel through the atmosphere constantly. Since there are regional inhomogeneities in the atmosphere's temperature, humidity, and pressure, the atmospheric phase is obtained by estimation rather than measurement.

*2.2. Phase Distribution Maximum Likelihood Estimation*

MLE is a statistical method for estimating model parameters. In the context of phase distribution parameters, MLE aims to find parameter values that maximize the probability of the phase distribution for a given observation. In deformation monitoring in low-coherence areas, the phase information of the observation points conforms to a specific probability distribution. The distribution parameters can reflect the stability and reliability of observation points, which are our preliminary screening criteria. For the target point $p$, the phase difference between two adjacent images is expressed as follows:

$$\mathbf{x}(\phi_p) = \left[ \phi_p(2) - \phi_p(1), \phi_p(3) - \phi_p(2), \ldots, \phi_p(n) - \phi_p(n-1) \right]. \tag{8}$$

Assuming the phase data follow a normal distribution, the probability density function of the normal distribution is

$$f(\mathbf{x}|\boldsymbol{\mu}, \Sigma) = \frac{1}{\sqrt{(2\pi)^k|\Sigma|}} \exp\left[ -\frac{1}{2}(\mathbf{x} - \boldsymbol{\mu})^\top \Sigma^{-1}(\mathbf{x} - \boldsymbol{\mu}) \right], \tag{9}$$

where $\mathbf{x}$ is a $k$-dimensional vector, $\mu$ is the mean vector, $\Sigma$ is the covariance matrix, and $|\Sigma|$ is the determinant of the covariance matrix. For a sample set $\mathbf{x} = \{\mathbf{x}_1, \mathbf{x}_2, \ldots, \mathbf{x}_n\}$, the likelihood function is

$$L(\mu, \Sigma | \mathbf{x}) = \prod_{i=1}^{n} f(\mathbf{x}_i | \mu, \Sigma). \tag{10}$$

Taking logarithms does not change the position of the extremes. For ease of calculation, taking both ends of the above equation logarithmically at the same time,

$$\ln L(\mu, \Sigma | \mathbf{x}) = \sum_{i=1}^{n} \ln f(\mathbf{x}_i | \mu, \Sigma) = \sum_{i=1}^{n} \left[ -\frac{k}{2} \ln(2\pi) - \frac{1}{2} \ln |\Sigma| - \frac{1}{2} (\mathbf{x}_i - \mu)^{\top} \Sigma^{-1} (\mathbf{x}_i - \mu) \right]. \tag{11}$$

We take partial derivatives with respect to $\mu$:

$$\frac{\partial}{\partial \mu} \ln L(\mu, \Sigma | \mathbf{x}) = \sum_{i=1}^{n} \Sigma^{-1} (\mathbf{x}_i - \mu). \tag{12}$$

Letting this partial derivative be zero, the equation is as follows:

$$\sum_{i=1}^{n} \Sigma^{-1} (\mathbf{x}_i - \mu) = 0. \tag{13}$$

The estimate is replaced by $\bar{\mu}$, then,

$$\bar{\mu} = \frac{1}{n} \sum_{i=1}^{n} \mathbf{x}_i. \tag{14}$$

The maximum likelihood estimate of $\bar{\mu}$ is the mean of the sample. We take partial derivatives with respect to $\Sigma$:

$$\frac{\partial}{\partial \Sigma} \ln L(\mu, \Sigma) = -\frac{n}{2} \Sigma^{-1} + \frac{1}{2} \Sigma^{-1} \left( \sum_{i=1}^{n} (\mathbf{x}_i - \mu)(\mathbf{x}_i - \mu)^{\top} \right) \Sigma^{-1}. \tag{15}$$

Letting this partial derivative be zero, the equation is as follows:

$$-\frac{n}{2} \Sigma^{-1} + \frac{1}{2} \Sigma^{-1} \left( \sum_{i=1}^{n} (\mathbf{x}_i - \mu)(\mathbf{x}_i - \mu)^{\top} \right) \Sigma^{-1} = 0. \tag{16}$$

The estimate is replaced by $\bar{\Sigma}$, then,

$$\bar{\Sigma} = \frac{1}{n} \sum_{i=1}^{n} (\mathbf{x}_i - \mu)(\mathbf{x}_i - \mu)^{\top}. \tag{17}$$

The maximum likelihood estimate of $\bar{\Sigma}$ is the covariance matrix of the sample. Normal distribution based on MLE is a commonly used and effective method that performs especially well with large samples and satisfies the normality assumption.

*2.3. Wavelet Filter*

The wavelet denoising algorithm serves as an essential method by suppressing the noise to maximize the retention of the meaningful part of the signal. The method utilizes the wavelet transform's multi-scale property to decompose the noise-affected signal into several scale components. The wavelet transform is the inner product of the mother

wavelet function shifted by $b$ with the signal $f(t)$ to be analyzed at different scales $a$, with the expression

$$WF(a,b) = \int_{-\infty}^{+\infty} \varphi_{a,b}(t)dt. \tag{18}$$

The scale shift transformation function is

$$\varphi_{a,b}(t) = \frac{1}{\sqrt{a}}\varphi\left(\frac{t-b}{a}\right), a > 0, b \in R, \tag{19}$$

where $a$ is the scale factor with a range greater than 0. $b$ is the shift factor, which can be positive or negative. The act of stretching is accomplished by adjusting the magnitude of $a$, while a shift is achieved by altering the value of $b$. The $\varphi_{a,b}(t)$ is the mother wavelet or fundamental. In the time domain, it is a bandpass function with zero mean, expressed as

$$\int_{-\infty}^{+\infty} \varphi_{a,b}(t) = 0. \tag{20}$$

The signal is decomposed into sub-signals of different frequency ranges using a wavelet transform. Subsequently, an appropriate threshold is set according to the characteristics of the scale. The sub-signal components with amplitudes that exceed the threshold are retained as helpful information. Meanwhile, sub-signal components with amplitudes that are below the threshold are considered noise components to be suppressed or removed.

Wavelet filtering filters out noise by thresholding, where the most critical step is the threshold selection. Thresholding for wavelet filtering is generally divided into hard and soft thresholding. The soft-threshold expression is shown below:

$$s = \begin{cases} \text{sign}(x)(|x| - T), & |x| > T \\ 0, & |x| \leq T \end{cases}, \tag{21}$$

where $\text{sign}(x)$ denotes the positive or negative sign of $x$. $T$ is the threshold. $x$ is the wavelet coefficients after being decomposed. $s$ is the wavelet coefficients after threshold filtering. In the soft-threshold case, data exceeding the absolute value of the wavelet threshold become a subtraction of two values. In contrast, data less than the absolute value of the wavelet threshold become zero. The hard-threshold expression is shown below:

$$s = \begin{cases} x, & |x| > T \\ 0, & |x| \leq T \end{cases}. \tag{22}$$

In the hard-threshold case, the data are retained when the absolute value of the wavelet coefficients is greater than the threshold while becoming zero when the wavelet coefficients are less than the threshold. Relevant experimental studies [44] show that hard thresholding may produce discontinuities that lead to oscillations or artifacts in the reconstructed signal, while soft thresholding makes the reconstructed signal smooth. Therefore, we use soft thresholding to filter the phase signal.

The size of the threshold affects the filtering effect. Figure 2 shows a standard normal distribution curve with different confidence intervals labeled. The A area represents the 68.3% confidence interval (±1 standard deviation), the B area represents the 95.5% confidence interval (±2 standard deviation), and the C area represents the 99.7% confidence interval (±3 standard deviation). Typically, the threshold is set around $3\sigma$, which filters out 99.7% of the phase noise.

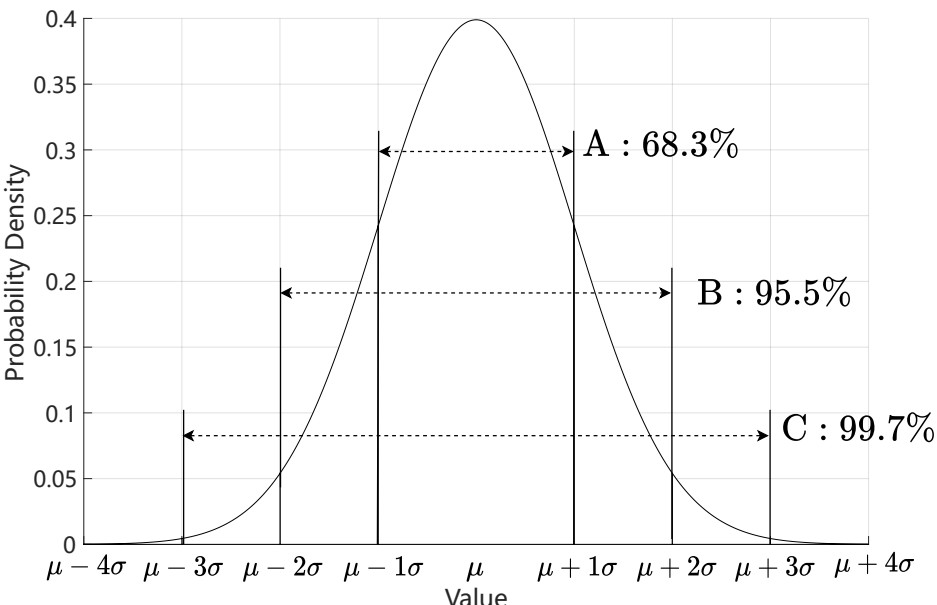

**Figure 2.** Normal distribution with mean $\mu$ and standard deviation $\sigma$. A areas indicate 68.3% confidence intervals, B areas indicate 95.5% confidence intervals, and C areas indicate 99.7% confidence intervals.

In deformation monitoring, there is both high-frequency and low-frequency information in the deformation monitoring phase. The period information, deformation trend, and atmospheric phase information are slowly changing, and the mutation information is rapidly changing. The deformation trend contains information about slow changes in terrain, which is essential for deformation monitoring. Wavelet filtering plays a filtering and smoothing role for noise signals or interference signals that are smaller than the threshold, while it plays a detection role for signals that are greater than the threshold.

## 3. Reliable Observation Point Selecting Process

In low-coherence scenarios, there is a high level of noise or mutation information in the target phase information. Mutation information often results in unwrapped phase errors, and phase unwrapped errors directly affect the results of deformation processing. Higher noise information affects phase stability. For the detection of low-coherence areas, this paper proposes a screening process for low-coherence areas, and the whole deformation information processing process is shown in Figure 3.

The whole signal processing contains 10 steps. In particular, the signal pre-processing mainly comprises step 1 to step 6, to improve the density of observation points. Step 7 and step 8 screen the observation points to improve the accuracy of the observation points. Step 9 and step 10 are atmospheric phase estimation and removal. The processing of each step is described in detail in Sections 3.1 and 3.2.

### 3.1. Data Acquisition and Pre-Processing

GB-SAR continuously transmits a Frequency Modulated Continuous Wave (FMCW) signal from a fixed ground position. These signals are transmitted toward the detection area, and the system records the reflected signal. The echo signals contain information reflected from different objects and ground features. These data are systematically calibrated and compensated to result in SAR images [9].

Step 1 is the MLE estimation of the differential phase information. Differential phase information is the phase difference between two neighboring images, which can effectively eliminate the effect of the initial phase or linear trend. The differential phase accomplishes

the operation in the complex domain, and the differential phase range is $-\pi$ and $\pi$. Phase mutations often occur in low-coherence areas with low signal-to-noise ratios, which are indicated by the appearance of higher amplitude pulses for a short time. The occurrence probability of the mutation phase is directly proportional to the standard deviation of the distribution function, a relationship corroborated by the posterior probability. The probability of a mutation phase diminishes with a decrease in the standard deviation of the phase information.

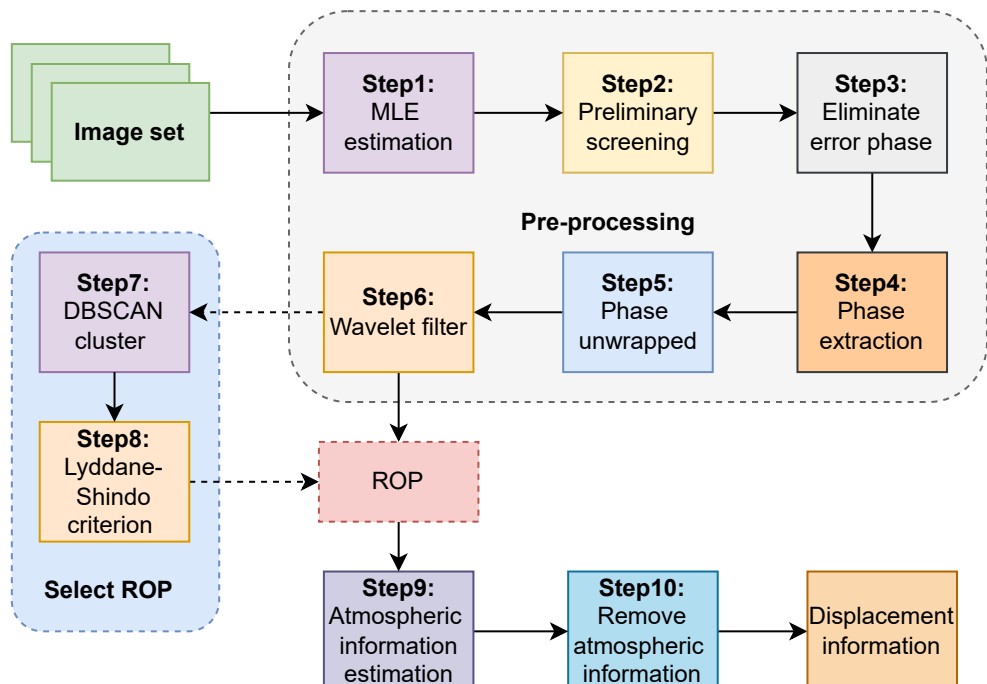

**Figure 3.** The proposed ROP selection method.

The estimation of the parameters of a Gaussian distribution is accomplished through maximum likelihood estimation, with the accuracy of the estimation improving as the sample size increases. The phase of stable observation points conforms to a Gaussian distribution. However, as noise increases, phases exceeding $\pm\pi$ are wrapped back within the range of $\pm\pi$, gradually approaching a uniform distribution. Assessing the variance can evaluate the stability of the phase. According to the literature [45], the critical variance for completely decorrelated points is $\pi^2/3$. Points with a variance greater than that value are considered to be noise.

Step 2 involves an initial selection of observation points, guided by estimated parameters. These parameters gauge the reliability of the observation points, with a lower standard deviation indicating higher confidence. Consequently, a reduced standard deviation results in a more selective screening process, potentially diminishing the number of observation points. Nonetheless, an excessively limited set of observation points may affect the extent of the observational coverage. Therefore, the screening threshold should be tailored to the specific observational context. Typically, scatterers with a standard deviation greater than 1.5 are meaningless for processing. This criterion strikes a balance between retaining an adequate quantity of observation points and incurring a certain level of phase error amongst them.

Step 3 is to eliminate the phase error or phase correction. Commonly used error correction methods are interpolation, median filtering, and mean filtering. The interpolation method is to interpolate the phase where the absolute value of the differential phase exceeds $\pi$. When filters are used, the length of the filter should be moderate, because a longer filter will filter out the useful deformation information.

Step 4 is the phase extraction of observation points, which involves converting complex information into phase information. The image set contains both magnitude and phase information. The magnitude information in a low-coherence scene does not fully reflect the stability of the phase, so the phase information is extracted for more accurate analysis. The following equation can represent the process of phase extraction:

$$\phi_{ex}(t_i) = \arg\{s(t_i)\}, i = 1, 2, \ldots, N, \tag{23}$$

where $\phi_{ex}(t_i)$ denotes the point phase information at moment $t_i$. $s(t_i)$ is the phase-corrected complex information. $arg\{\cdot\}$ denotes the phase calculation of complex data.

Step 5 is phase unwrapping, which recovers the actual phase of the target from the wrapped phase information. The range of phase information calculated in the phase extraction step is $(-\pi, \pi)$, as the exceeding phases are also folded back into that range. The following equation can represent the process of phase unwrapping:

$$\phi_{unwrapped}(t_i) = \phi_{ex}(t_i) + 2\pi k_i , \tag{24}$$

where $\phi_{unwrapped}(t_i)$ is the true phase at moment $t_i$ after phase unwrapping. $\phi_{ex}(t_i)$ is the wrapped phase information at that moment, calculated directly from the image data. $k_i$ is the unwrapping constant, which must be added during the unwrapping process to ensure that the phase values are continuous after the unwrapping. We judge the phases of two adjacent images to return the wrapped phase to the actual phase information. The judgment condition is

$$k_{i+1} = \begin{cases} k_i + 1 & , \phi_{ex}(t_{i+1}) - \phi_{ex}(t_i) < -\pi \\ k_i & , -\pi \le \phi_{ex}(t_{i+1}) - \phi_{ex}(t_i) \le \pi \\ k_i - 1 & , \phi_{ex}(t_{i+1}) - \phi_{ex}(t_i) > \pi \end{cases} . \tag{25}$$

The condition for the above equation to be true is that the motion of the target point is not ambiguous. In other words, the motion distance in the sampling interval cannot be greater than $\lambda/4$.

Step 6 involves a wavelet filter that adaptively filters observation point phases or deformation curves by adjusting thresholds based on MLE standard deviation. The screened observations have different estimated standard deviations due to their different stability. Proper thresholding filters out most of the noise and retains the deformation trend information. The specific realization is that the threshold value is the standard deviation multiplied by the weight factor. The threshold of the wavelet filter can be adjusted according to the MLE standard deviation so that less stable observation points can achieve the same smoothness as stable observation points.

### 3.2. Reliable Observation Point Screening and Atmospheric Phase Processing

Filtering can remove the noise between several nearby phases, but it cannot constrain the deformation trend of the observation points. Therefore we perform spatial dimension clustering filtering to select observation points with correct deformation trends. In step 7, the deformation data are clustered using the DBSCAN algorithm to identify the most consistent set of observation points, while rejecting outliers that do not belong to any group. These outliers are usually unreliable observations. Step 8 removes significant anomalies and large deformation magnitude observations from the reliable group based on the Lyddane–Shindo criterion. Step 9 fits the optimal atmospheric phase curve using the MMSE estimation method. Step 10 removes the atmospheric phase information to obtain the deformation information.

In step 7, the DBSCAN algorithm often identifies and eliminates outliers. Due to the variable stability of the observation points, the dataset may still contain outliers even after filtering. These anomalous data typically exhibit significant deviations from average data. As a density clustering tool, the DBSCAN algorithm effectively differentiates between different data groups and identifies ROPs and anomalies.

The DBSCAN algorithm is characterized by two key parameters: the neighborhood radius $\varepsilon$ and the minimum number of points $N_{Pmin}$. Let us assume a given dataset $D = \{x_1, x_2, \ldots, x_M\}$, where $M$ represents the number of deformation curves. The neighborhood radius $\varepsilon$ is defined as follows: for any point $x_j \in D$ in the dataset, a search radius within which the distance to $x_j$ is less than $\varepsilon$, i.e., $N_\varepsilon = \{x_j \in D | dis(x_i, x_j) \leq \varepsilon\}$. If the neighborhood number of $x_j$ is greater than $N_{Pmin}$, denoted as $N_\varepsilon(x_j) \geq N_{Pmin}$, then $x_j$ is defined as a core point. If a point $x_j$ in the $\varepsilon$ neighborhood of $x_i$ and $x_i$ is a core point, then $x_j$ is density-reachable with $x_i$. Density-reachable exhibits transitivity, e.g., if core points $x_j$ and $x_i$ are not within each other's neighborhood radius, but both $x_j$ and $x_i$ are within the neighborhood radius of $x_k$, then $x_j$ and $x_i$ are density-reachable.

Based on the core concept of the DBSCAN algorithm, a point $p$ is randomly chosen as the starting point in the dataset $D$. Then, with the specified parameters $\varepsilon$ and $N_{Pmin}$, we find all the points that can be reached from the point $p$ density. If $p$ represents a core point, all points within its $\varepsilon$ neighborhood form a cluster. Next, the density reachability of these points is examined to extend this cluster further until the complete set is identified. If $p$ is not a core point, the search moves to the next point until all points in the dataset have been evaluated. Eventually, points in the dataset that do not belong to any cluster are classified as outliers. Curves containing outliers are considered unreliable observations.

In step 8, the Lyddane–Shindo criterion uses the triple standard deviation $\sigma$ of observations as the limit trade-off standard. Therefore, the Lyddane–Shindo criterion is also known as the $3\sigma$ criterion. The standard deviation is a parameter calculated after extensive repeated observations.

The basis for judgment under the Lyddane–Shindo criterion is as follows:

$$e_{x_m} = |x_m - \bar{x}|, \tag{26}$$

where $e_{x_m}$ represents the residual in the formula. In mathematical statistics, the residual refers to the difference between the actual observed value and the estimated value, which is the observed value of the error. Residuals greater than $3\sigma$ are significant errors that should be compensated for or discarded. On the contrary, if the residual is less than or equal to $3\sigma$, it is classified as normal data that should be retained.

Generally, the Lyddane–Shindo criterion is applicable in scenarios involving many observations, as its accuracy improves with extensive data calculation. According to probability statistics, when data follow a normal distribution, the probability of residuals exceeding $3\sigma$ is exceedingly tiny. Therefore, most instability points are screened when the Lyddane–Shindo criterion discriminates outliers.

In step 9, the MMSE is a commonly used estimation method in signal processing. It aims to find an estimator that minimizes the Mean Square Error (MSE) between the estimated and actual values. The MSE is the expected value of the square of the estimation error, commonly used to measure the accuracy of an estimated quantity. Mathematically, if $\widehat{\theta}$ is the estimator of $\theta$, then the mean square error (MSE) is defined as

$$\text{MSE}\left(\widehat{\theta}\right) = \text{E}\left[\left(\widehat{\theta} - \theta\right)^2\right] \tag{27}$$

where E denotes the expected value. The minimum Mean Square Error aims to find an estimator $\theta$ that minimizes the MSE. Optimization algorithms such as the gradient descent method achieve minimum mean square error estimation.

## 4. Experimental Results and Analysis

To evaluate the effectiveness of the proposed method in GB-SAR deformation monitoring, this experiment analyzed the data measurements in a vegetation-covered scenario. The experimental scenario is shown in Figure 4. Figure 4a is an optical picture, and Figure 4b is a SAR image. In the SAR image, the scene targets in the red rectangle have shallow target scattering coefficients at a distance of 200 to 280 m with relatively poor coherence due to vegetation. The experiments are divided into two parts: the first part involves selecting

reliable observation points using the proposed method; the second part tests the accuracy of the proposed method.

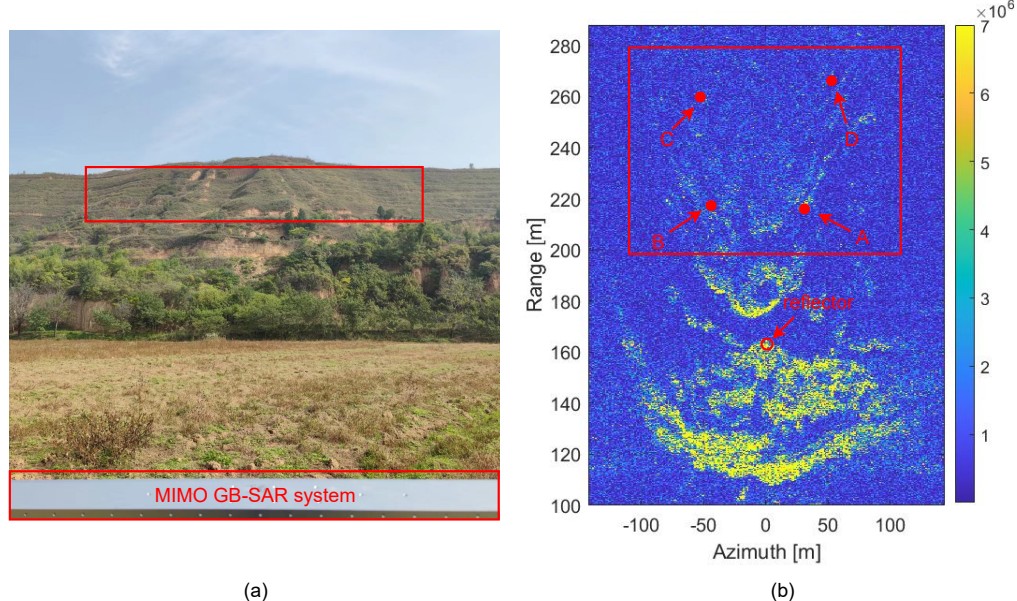

|             (a)             |          (b)          |

**Figure 4.** Experimental scenarios with vegetation cover. (**a**) The view of the GB-SAR system. (**b**) SAR image results.

For deformation monitoring, a high-speed GB-SAR system [9] was used with the parameters shown in Table 1. The image acquisition rate can be as high as 100 frames per second. The image acquisition rate can be flexibly adjusted according to different application scenarios. In landslide scenarios, the image sampling rate was adjusted from one frame per second to one frame per few minutes. The first part of the experiment was 3.3 h long, with an observation interval of 24 s, and the number of images acquired was 500. The second part of the experiment was 4.6 hours long, the observation interval was 24 s, and the number of images acquired was 700.

**Table 1.** GB-SAR system parameters.

| Items | Value |
|---|---|
| Center frequency | 30 GHz |
| Frequency band | 1000 MHz |
| ADC sampling rate | 400 MSPS |
| Single ramp time $T$ | $\geq$20 μs |
| Time for a single full scan | $\geq$4.96 ms |
| Detection distance | 20–2000 m |

### 4.1. Data Pre-Processing

Data pre-processing mainly comprises three parts. The first part is the parameter estimation of the phase distribution based on MLE, which is mainly used to obtain the distribution model through estimation. The second part is the mutation information correction, which mainly addresses the unwrapping phase error. The third part is mainly wavelet filtering, which filters the noise using the estimated distribution parameter.

To verify that the phase distribution conforms to a Gaussian distribution, MLE was performed for a number of points from A to D in Figure 4, with the estimation outcomes presented in Figure 5. The bar graph represents the actual statistical measurements, while the red curve depicts the estimated probability distribution curve. Specifically, Figure 5a illustrates the estimation results for point A, exhibiting a standard deviation of approximately 0.5. Figure 5b details the results for point B, where the standard deviation stands at 1.0.

Figure 5c displays the findings for point C, with a standard deviation of 1.4, and Figure 5d outlines the estimations for point D, showing a standard deviation of 1.8. The collective results from Figure 5a–d affirm that, for low-noise observations, the phase conforms to a normal distribution, gradually converging to a uniform distribution with a probability distribution function of $1/2\pi$ as the noise increases.

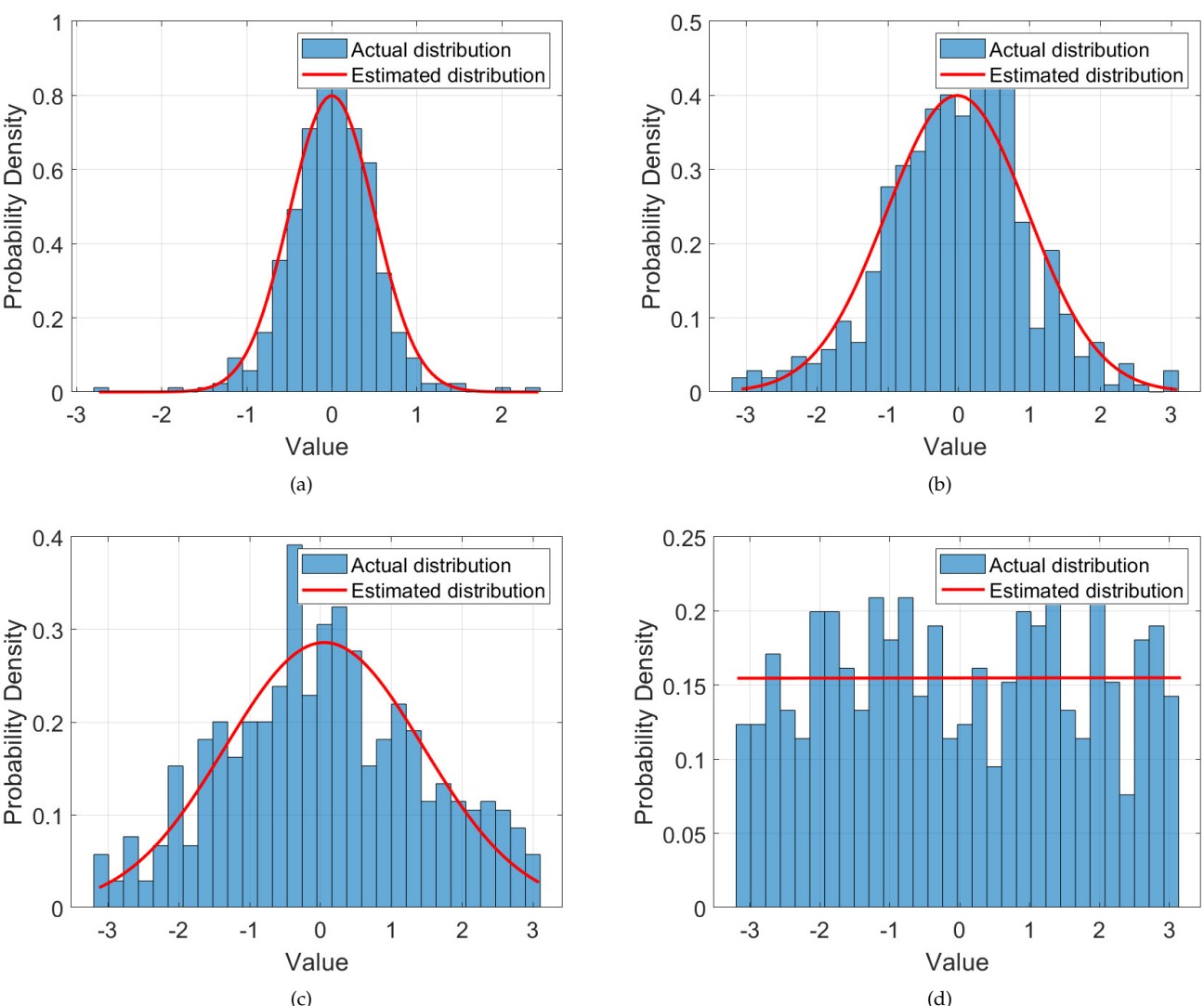

**Figure 5.** The maximum likelihood estimate of the phase distribution parametric model (**a**) with standard deviation 0.5 and mean 0, (**b**) with standard deviation 1.0 and mean 0, (**c**) with standard deviation 1.4 and mean 0, and (**d**) with standard deviation 1.8 and mean 0.

Figure 6 presents the differential phase information for the corresponding distribution parameters. Concurrently, Figure 7 exhibits the unwrapped phase information corresponding to these parameters. In Figure 6a, the curve's standard deviation is 0.5, showcasing that most differential phases are confined within the $[-1.5, 1.5]$ radian range. Correspondingly, Figure 7a reveals stable unwrapped phase information without errors in phase unwrapping. Figure 6b illustrates that the curve standard deviation is 1.0 and the phase is mostly distributed in the range $[-2, 2]$ radian, which is relatively stable. Figure 7b displays the corresponding unwrapped phase information with one unwrapped error. In Figure 6c, the curve standard deviation increases to 1.4, as evidenced by more differential phases approaching $\pi$, suggesting a decrease in phase stability. This is further corroborated in Figure 7c, where the unwrapped phase information contains a dozen unwrapping errors. The unwrapped phase information shows a stepped pattern, due to the presence of un-

wrapped errors. Figure 6d displays a curve with a standard deviation of 1.8, indicating poor phase stability. Figure 7d shows the differential phase information for the incoherent points, where the unwrapped phase information shows randomness. The progression from Figures 5–7 clearly indicates a decline in phase information stability as standard deviation rises, alongside an escalated likelihood of unwrapped phase errors.

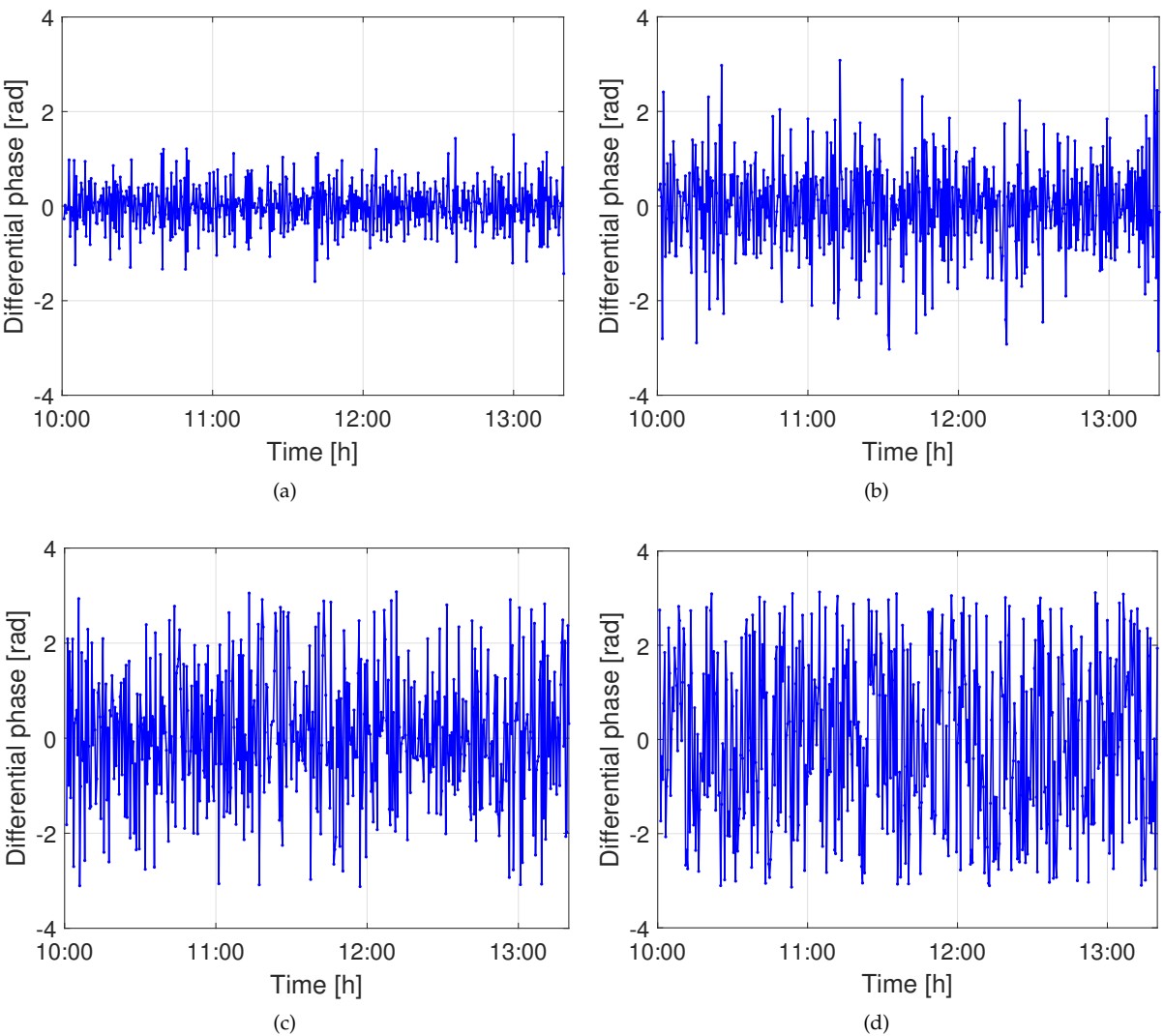

**Figure 6.** Differential phase information at different standard deviations (**a**) with standard deviation 0.5, (**b**) with standard deviation 1.0, (**c**) with standard deviation 1.4, and (**d**) with standard deviation 1.8.

The red curve in Figure 8 contains multiple unwrapped phase errors. Unwrapped phase errors show more pronounced phase jumps that deviate from the standard curve $2\pi$ with random jump directions. This is caused by a relatively large phase mutation, and two consecutive relatively large noises cause unwrapped phase errors. The blue curve shows the corrected phase information.

The mean and median filters can filter out the larger noise, and the minor noise is still noticeable. To further filter the noise, the phase information is wavelet filtered. The red curve in Figure 9 shows the curve before filtering, and the noise is more evident with an amplitude of 2–3 mm. The blue curve is the deformation curve after filtering, which is relatively smooth. Wavelet filtering can be used to detect larger deformation information, such as the deformation in the circle in Figure 9.

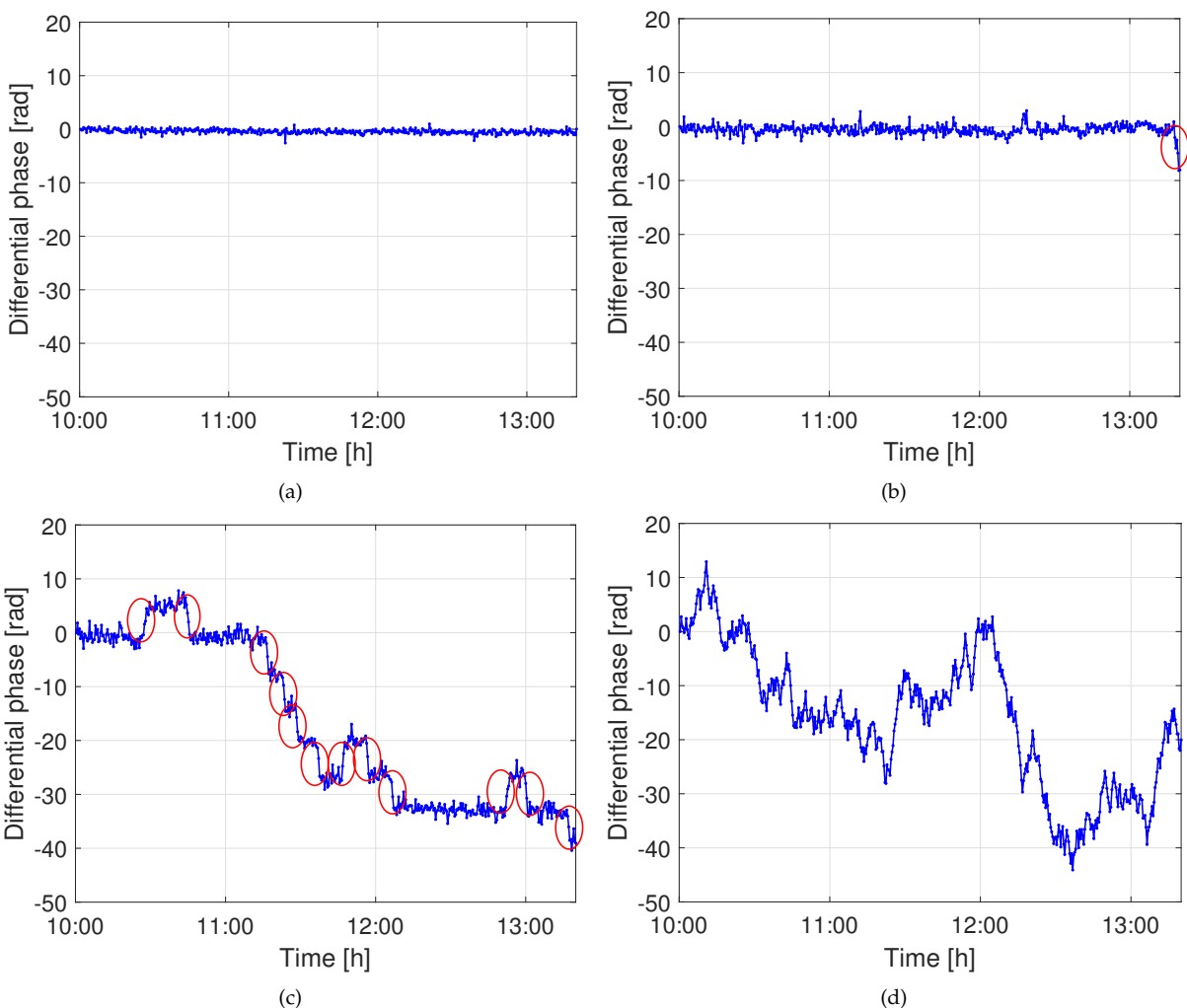

**Figure 7.** Phase unwrapping information at different standard deviations (**a**) with standard deviation 0.5, (**b**) with standard deviation 1.0, (**c**) with standard deviation 1.4, and (**d**) with standard deviation 1.8. Red circles indicate unwrapped phase errors.

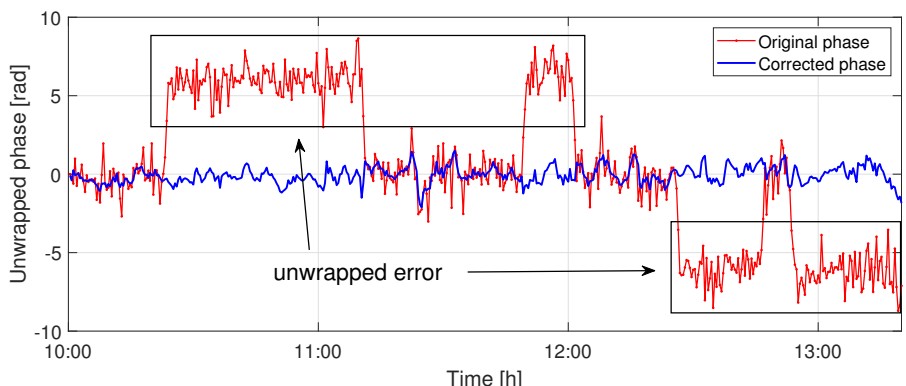

**Figure 8.** Unwrapped phase error and corrected phase. The red generation contains the unwrapped phase error, and the blue is the corrected phase.

Figure 10 displays the comparative results after different processing steps. Figure 10a shows the result without phase correction, where red points represent unstable scatterers. In Figure 10b, the observation points in Figure 10a with deformations less than 10 mm are screened, totaling 5084 points. Figure 10c presents the outcome after phase correction,

where the density of candidate points significantly increased from 5084 to 31,020 points. Figure 10d illustrates the result following a wavelet filtering step. Compared to the unfiltered result, the phase stability is improved and deformation is reduced after filtering, hence the deeper blue color. After data pre-processing, both the density and phase stability of observation points have markedly improved.

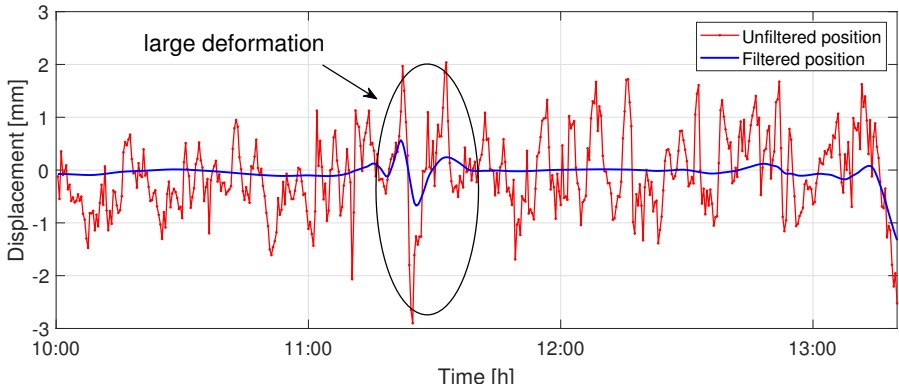

**Figure 9.** Example of wavelet filtering results. Red represents the curve before filtering, and blue represents the curve after filtering.

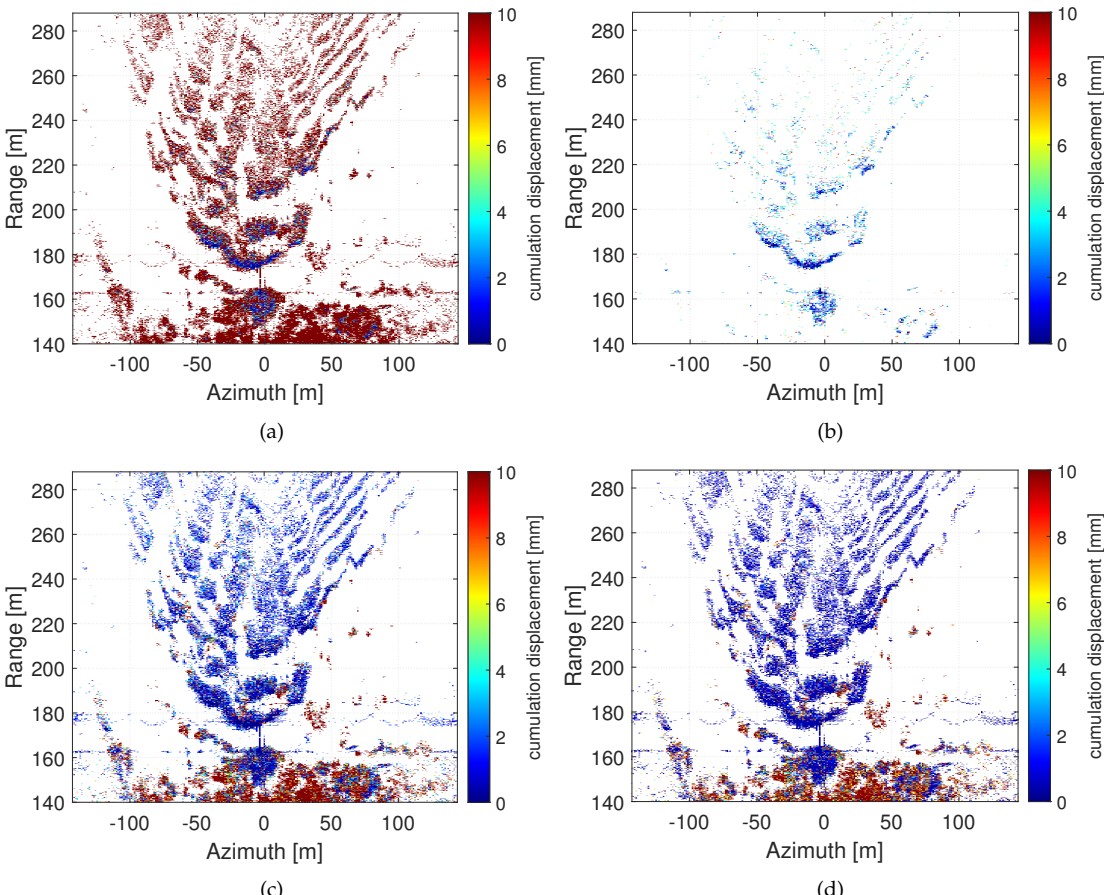

**Figure 10.** The comparison results at different processing steps. (**a**) is the processing result before phase correction, with standard deviation less than 1.5. (**b**) is the observation point with deformation lower than 10 mm in (**a**). (**c**) is the result after phase correction, with the same standard deviation as (**a**). (**d**) is the processing result after filtering.

### 4.2. Clustering–Screening and Atmospheric Phase Estimation

The filtered target points contain both correct and incorrect deformation trends. The clustering–screening method selects the ROPs with correct deformation trends. The selected ROP curves are used to estimate the atmospheric phase. To verify the effectiveness of the clustering–screening and estimation method, the deformation data are first clustered using the DBSCAN algorithm, and the clustered data are screened using the Lyddane–Shindo criterion. Atmospheric phase estimates are obtained by MMSE fitting based on screened curves.

Figure 11 shows the deformation curves for part of the observations in Figure 10d. The blue curves are ROPs filtered by DBSCAN clustering combined with the Lyddane–Shindo criterion. The blue observation curve area has the same density, and the curve is the most stable. Red deformation curves indicate that not all points are within the $3\sigma$ limits or that some curves have a deformation trend that is separated from the majority of the deformation curve. Red curves are considered to be un-ROP.

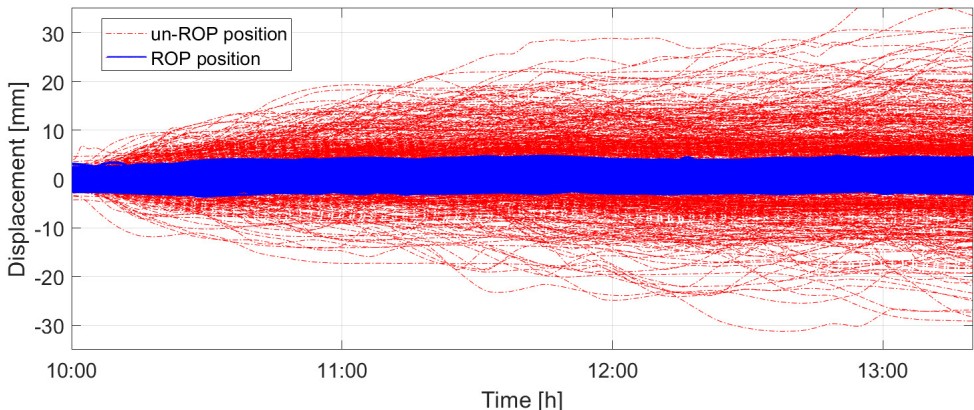

**Figure 11.** Results of clustering–screening. Red indicates un-ROP curves, and blue indicates ROP curves.

Figure 12 shows the changes in the MMSE estimation curves before and after removing these un-ROP. In the figure, the red curves indicate the MMSE estimation results for scene data containing stable and unstable curves, and the blue curve represents the ROP curve estimation results after screening. The difference between the fitting results before and after the cluster screening gradually increases is caused by the deviation of the instability point from the normal deformation trend. The pink curve represents the deformation curve of the corner reflector in Figure 4. With no deformation, the deformation curve of the corner reflector is the atmospheric phase curve. It is proved that after removing the observation points with incorrect deformation trends, the estimated curves of the reliable observation points are closer to the actual deformation curves.

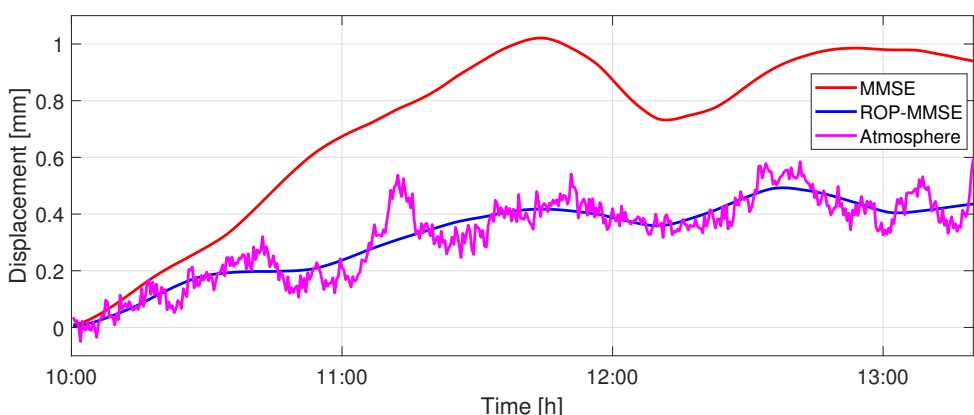

**Figure 12.** Comparison results of MMSE curve estimation before and after clustering–screening.

Figure 13a,b show that there is a significant difference before and after clustering screening. Figure 13a shows a large blue area with a certain amount of red dots mixed in, especially at close distances where the red dots are more pronounced. Figure 13b shows only a patch of blue points. Before the clustering screening step, the observation area contains mostly points with regular deformation trends and a small number of irregular points, and after the screening, there are only observations with regular deformation trends.

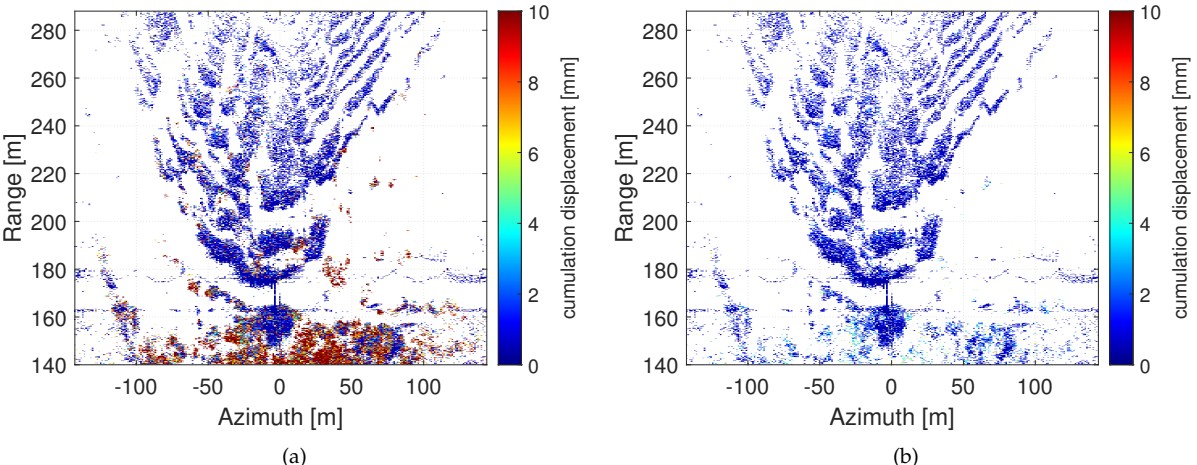

(a)                                                    (b)

**Figure 13.** Processing results (**a**) before cluster screening and (**b**) after cluster screening.

This proves the successful application of the clustering–screening method in deformation processing. Since curves with large changes in volatility trends also have some weight in the curve estimation process, this can cause deviations in the estimation results. Through clustering–screening, these observations with fluctuating deformation trends were effectively identified and excluded, and the accuracy and reliability of the fitted curves were improved. The significance of this is that, without knowing the deformation distribution and trend of the observation points, the clustering–screening can separate the reliable and unreliable observation points.

*4.3. Comparison with Amplitude Deviation PS Method*

In the second part of the experiment, the radar's position remained stationary while continuous observation of the area was conducted. To differentiate from the first part of the experiments, artificial deformation was introduced in the second experiment. Both the amplitude deviation PS method and the proposed method were employed for processing. While the selection methods differed, data pre-processing remained the same for both approaches. The selection of reliable observation points for the proposed method was determined by the first part of the experiment, and the observation points were not screened in this experiment.

Figure 14 presents the image outcomes from both selection methods. Deformations identified by each method are highlighted with red circles in Figure 14, and the resulting deformation curves align due to the application of identical processing steps across both approaches. However, the precision in pinpointing reliable observation points diverges between the two methods. Figure 14a shows the target points selected using an amplitude deviation method with a threshold value of 0.25, while Figure 14b utilizes the proposed method selecting the same number of 846 ROPs. While both methods achieve high accuracy and similar observation distributions, the number of observed results they produce is quite limited. Figure 14c,d feature the same number of 8864 ROPs, with a standard deviation threshold of 1.0. Figure 14d demonstrates better accuracy in identifying reliable points in local areas compared to the amplitude dispersion method, though the difference is slight. The processing result in Figure 14f was achieved using the proposed method with

a standard deviation threshold of less than 1.5. Figure 14e,f share the same 27,707 ROPs. Figure 14f exhibits a higher accuracy in identifying reliable points compared to Figure 14e.

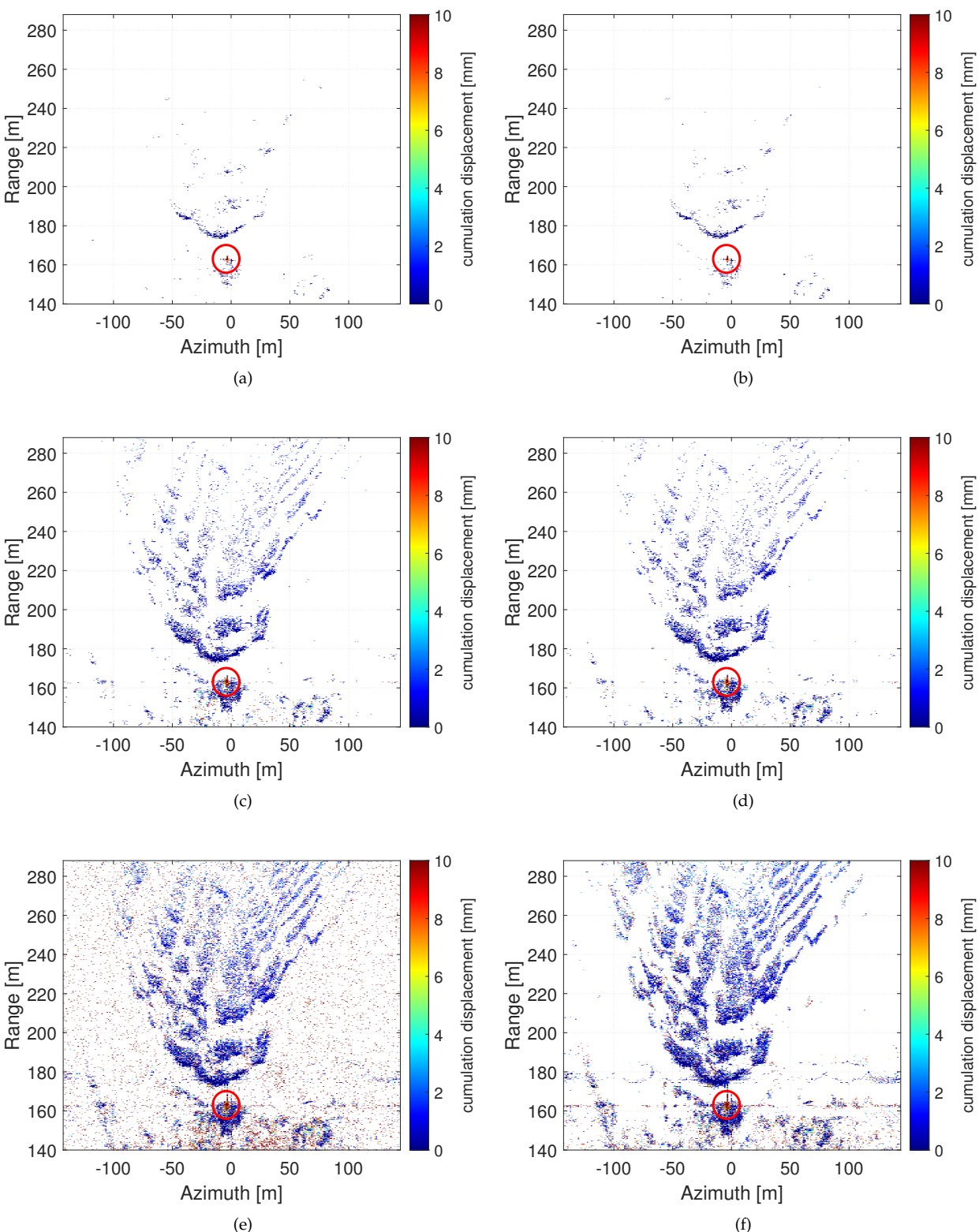

**Figure 14.** Comparison of the proposed ROP method with the conventional method. The amplitude deviation PS processing results (**a**,**c**,**e**). The proposed ROP method processing results (**b**,**d**,**f**). Red circles indicate areas of artificial deformation.

Figure 15 presents the accuracy results of two methods. It demonstrates that the proposed method outperforms the amplitude deviation selection method in terms of accuracy. Although the accuracy of target points decreases with an increase in the standard deviation of estimates, the advantage of the proposed method grows as the number of target points increases. Table 2 shows a data comparison between the proposed method and the amplitude deviation PS method. As the phase standard deviation increases, the filtering threshold for the amplitude deviation method also rises. When the phase standard deviation of the proposed method reaches 1.5, the amplitude standard deviation nears 0.5, where the amplitude standard deviation cannot fully reflect the stability of the phase standard deviation. In Table 2, a significant drop in accuracy is observed for a standard deviation greater than 1.5, which is attributed to the insufficient number of reliable observation points in the target area. With the screening condition of a standard deviation of 1.5, the accuracy of stable observation points can be improved by 14.66%.

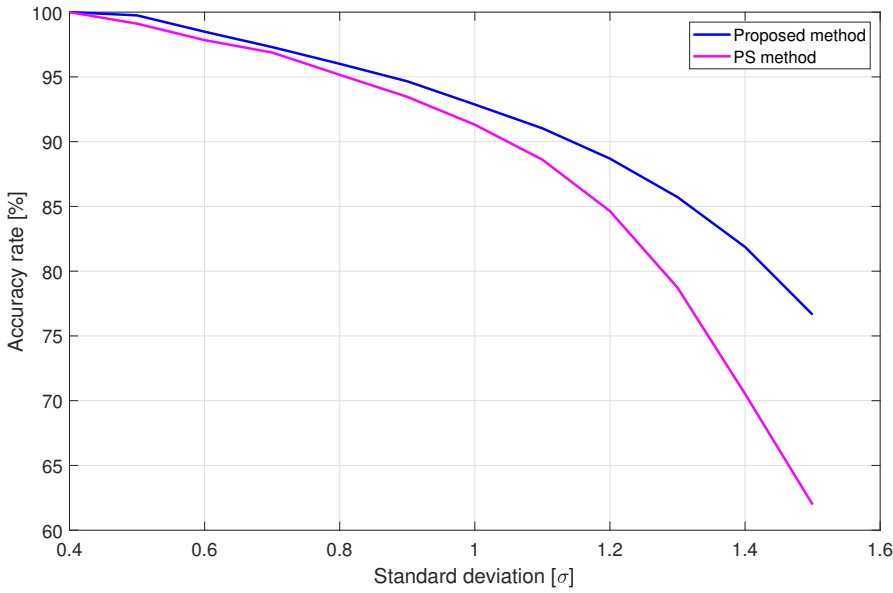

**Figure 15.** Comparison results of MMSE curve estimation before and after clustering–screening.

**Table 2.** The relationship between different estimation standard deviations and accuracy.

| Standard Deviation | Total ROP Number | Amplitude PS Method Accuracy Rate [%] | Proposed Method Accuracy Rate [%] | Amplitude Dispersion Index |
|---|---|---|---|---|
| $\sigma = 0.4$ | 1222 | 100.0 | 100.0 | 0.278 |
| $\sigma = 0.5$ | 2024 | 99.75 | 99.11 | 0.319 |
| $\sigma = 0.6$ | 2912 | 98.49 | 97.84 | 0.350 |
| $\sigma = 0.7$ | 4005 | 97.30 | 96.88 | 0.378 |
| $\sigma = 0.8$ | 5309 | 96.01 | 95.16 | 0.402 |
| $\sigma = 0.9$ | 6869 | 94.66 | 93.46 | 0.423 |
| $\sigma = 1.0$ | 8864 | 92.87 | 91.31 | 0.441 |
| $\sigma = 1.1$ | 11,155 | 91.03 | 88.61 | 0.456 |
| $\sigma = 1.2$ | 14,022 | 88.69 | 84.64 | 0.469 |
| $\sigma = 1.3$ | 17,526 | 85.72 | 78.73 | 0.478 |
| $\sigma = 1.4$ | 22,098 | 81.87 | 70.53 | 0.485 |
| $\sigma = 1.5$ | 27,707 | 76.64 | 61.98 | 0.491 |
| $\sigma = 1.6$ | 45,718 | 55.07 | 44.73 | 0.501 |
| $\sigma = 1.7$ | 78,070 | 36.10 | 30.90 | 0.510 |
| $\sigma = 1.8$ | 162,860 | 17.91 | 19.25 | 0.531 |

## 5. Conclusions

Phase correction and the correct selection of deformation trends are crucial for deformation detection in areas of low coherence. In this paper, by applying phase correction to observation points and combining this with a clustering filtering method, the number of reliable observation points increased from 5084 (before correction) to 27,707, achieving an approximate fourfold increase. This process significantly improved the density of observation points, providing robust data support for the accuracy of deformation detection. Moreover, experimental validation in real scenarios revealed that using a standard amplitude deviation value of 0.25 to select stable observation points resulted in a limited number, leading to suboptimal regional detection effects. By relaxing the selection criteria to an amplitude deviation value of 0.49, a larger number of observation points could be obtained. However, the standard amplitude deviation PS method does not include the phase correction and other data pre-processing steps proposed in this paper. Without these pre-processing steps, the accuracy of the amplitude deviation PS method would be lower than 18.35%. By adopting the proposed method, the accuracy reached 76.64%, effectively enhancing the precision of ROP selection.

For high coherence areas, such as urban or structural environments, high signal-to-noise ratio amplitude deviations can accurately reflect phase deviations, making the amplitude deviation PS method an appropriate choice for selecting observation points. However, in non-urban environments or vegetated areas, due to weaker scattering characteristics and lower coherence, amplitude deviations no longer effectively reflect phase deviations, and phase information is unstable. In these cases, the proposed method effectively solves the problem of identifying observation points, making it a preferable option. Finally, theoretical analysis and experimental results validated the effectiveness of the proposed method for selecting ROPs in low-coherence areas.

**Author Contributions:** Conceptualization, Z.Z. and Z.L.; methodology, Z.Z., Z.L. and Z.S.; software, Z.Z. and L.Q.; validation, Z.Z.; formal analysis, Z.Z. and Z.L.; investigation, Z.Z.; resources, Z.L. and Z.S.; data curation, Z.Z.; writing—original draft preparation, Z.Z. and L.Q.; writing—review and editing, Z.L., F.T., H.G. and H.T.; visualization, Z.Z.; supervision, Z.L., Z.S. and H.T.; project administration, Z.Z. and Z.L.; funding acquisition, Z.L. and Z.S. All authors have read and agreed to the published version of the manuscript.

**Funding:** This research was funded by the National Natural Science Foundation of China under grant number 62031005.

**Data Availability Statement:** Data are contained within the article.

**Acknowledgments:** The authors gratefully acknowledge the helpful comments and suggestions of the reviewers of this manuscript.

**Conflicts of Interest:** The authors declare no conflicts of interest.

## Abbreviations

The following abbreviations are used in this manuscript:

| | |
|---|---|
| ADC | Analog-to-Digital Converter |
| DBSCAN | Density-Based Spatial Clustering of Applications with Noise |
| FMCW | Frequency Modulated Continuous Wave |
| GB-SAR | Ground-Based Synthetic Aperture Radar |
| MIMO | Multiple Input Multiple Output |
| MLE | Maximum Likelihood Estimation |
| MMSE | Minimum Mean Square Error |
| MSE | Mean Square Error |
| PS | Permanent Scatterer |
| ROP | Reliable Observation Point |
| SAR | Synthetic Aperture Radar |

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
