# Peer review of "A Reliable Observation Point Selection Method for GB-SAR in Low-Coherence Areas"

_remotesensing, doi:10.3390/rs16071251_

Round 1

Reviewer 1 Report

Comments and Suggestions for Authors

Dear authors,

Thank you for submitting your manuscript. I have carefully reviewed your work and find it to be a valuable contribution to the field of GBSAR data processing for deformation signal analysis in low-coherence areas. Your proposed framework, which includes phase distribution exploration, phase stability analysis, phase correction, and phase filtering, shows promise for practical applications, particularly in interpreting deformation products in mountain areas. Overall, the structure of the manuscript is well organized, and the content is well-written. It is evident that you have conducted thorough research, as demonstrated by the final results. The performance and effectiveness of your proposed framework are commendable. My main concern is the innovation of the paper, as you may not highlight your innovation properly. On the other hand, your presentation skills should improve a lot. I have a few suggestions to improve the quality of the manuscript.

1. Title: Please consider revising the title to include keywords that accurately reflect the innovation and methodology of your approach.

2. Abstract and Conclusion: Both the abstract and conclusion sections need improvement. Please ensure that the abstract provides a clear and concise statement of your whole study, highlighting the novelty and significance of your method. The conclusion section should provide a comprehensive summary of your findings and their implications.

3. Methodology: The methodology section should be redesigned to enhance its clarity and conciseness. Make sure to provide sufficient details about each step of your framework while avoiding unnecessary repetition.

4. Additional Experiments: It would be beneficial to include additional experiments or validation studies to further demonstrate the correctness and effectiveness of your phase correction method. This would strengthen the credibility of your proposed framework.

Comments on the Quality of English Language

No comments. The presentation skill should be improved a lot

Reviewer 2 Report

Comments and Suggestions for Authors

The paper presents a method for obtaining deformation time-deformation in low-coherence areas. Although interesting, I am afraid that the paper is not ready for publication. Several aspects are confusing and even some flaws present are too critical.

The paper is sometimes confusing and it is difficult to follow the rationale of the different steps of the algorithm. I believe it deserves a better explanation that guides the reader through the different steps.

At the beginning of section 2, the introduction of what is a GB-SAR is extremely poor. Not a single mention of the concept of Synthetic Aperture Radar, etc. It has to be improved.

The authors propose a quadratic model plus a stational term for the deformation. How well does this model match typical real deformations? Is it really necessary to rely on a model considering that the algorithm performs a phase unwrapping of the interferograms?

The authors are doing a temporal phase unwrapping of each pixel. Have the authors tried to combine it with a spatial one or even just using a spatial one? It would help to overcome the problems of temporally wrapped phases.

The whole processing of the data resides in the interferometric phase, but neither a single interferogram nor coherence map is shown. It would be interesting for the reader to see some of them to visualize the phase quality before and after the filtering, how noisy they are, if any deformation trend is visible, etc.

Figure 4 shows the phase distribution with different variances. The phase of an interferogram (or any measured phase like the one of an SLC) will show a phase distribution Gaussian only for low noise levels, but when the phase noise increases it will tend to a uniform distribution, as the phase is limited by the plus/minus pi bounds. This is why the interferometric coherence is a biased estimator. Are the results presented in the figure from a simulation or real data? If the data are simulated, the simulations are not correct.

Similarly, Figure 5 shows differential phase information plots at different variances. If the phase is wrapped, how can you have values over plus/minus pi? This is impossible. When operating with phases, any addition/subtraction operation has to be done in the complex domain. As the next step you are presenting is the unwrapping results, I guess you were working with wrapped ones in Figures 4 and 5.

It is not clear if the authors are processing real data or simulated one using as a base the image shown in Figure 3. There are no details regarding the observation period, number of images, if any ground truth was available, etc.

In my opinion, the paper presents a strong limitation regarding first the processing/simulation and, secondly, the validation of the method with real data. It is not clear if the authors work with real or simulated data and, from the results, one may deduce that the scene is perfectly stable. In this case, as your goal is to obtain an interferometric phase as constant as possible, which can be done in many ways, even a method that sets all phases to zero would produce a good result. Validation of an algorithm focused on obtaining deformation time-series with stable scenarios is quite unrealistic. In addition, if the scene is completely stable the model plays no role in the processing beyond evaluating that it did not induce processing instabilities.

Round 2

Reviewer 1 Report

Comments and Suggestions for Authors

Dear Author,

Thanks for taking the effort to address my comments. The revision is much better than before. Most of my comments have been addressed. The methodology and the experiments are described very clearly.

But in my opinion, could you please add some numerical value to highlight your improvements, such as how much accuracy you improved and how many points have been identified with your method compared to the traditional one.

On the other hand, the conclusion part should be rewritten; please identify how to write the conclusion for a paper. 

I have no more comments anymore.

Comments on the Quality of English Language

N/A

Author Response

Dear Reviewer,

First and foremost, I would like to express my deepest gratitude on behalf of my team for your willingness to dedicate your valuable time and effort to review our manuscript. Your positive evaluation and endorsement have been incredibly encouraging for us, and your insightful suggestions and valuable feedback have played a crucial role in enhancing our research work.

In response to your comments, we have incorporated numerical data in the experimental results section, highlighted in blue in the manuscript, to demonstrate the enhancements brought about by our method. We have thoroughly revised the conclusion part to, on one hand, quantify the differences between the two methods, thereby underscoring our advantages. On the other hand, we analyzed the applicability conditions of both methods, emphasizing the adaptability of our method in areas of low coherence. The revised conclusion is as follows:

“Phase correction and the correct selection of deformation trends are crucial for deformation detection in areas of low coherence. In this paper, by applying phase correction to observation points and combining this with a clustering filtering method, the number of reliable observation points increased from 5084 (before correction) to 27707, achieving an approximate fourfold increase. This process significantly improved the density of observation points, providing robust data support for the accuracy of deformation detection. Moreover, experimental validation in real scenarios revealed that using a standard amplitude deviation value of 0.25 to select stable observation points resulted in a limited number, leading to suboptimal regional detection effects. By relaxing the selection criteria to an amplitude deviation value of 0.49, a larger number of observation points could be obtained. However, the standard amplitude deviation PS method does not include the phase correction and other data preprocessing steps proposed in this paper. Without these preprocessing steps, the accuracy of the amplitude deviation PS method would be lower than 18.35%. By adopting the proposed method, the accuracy reached 76.64%, effectively enhancing the precision of ROP selection.

For high coherence areas, such as urban or structural environments, high signal-to-noise ratio amplitude deviations can accurately reflect phase deviations, making the amplitude deviation PS method an appropriate choice for selecting observation points. However, in non-urban environments or vegetated areas, due to weaker scattering characteristics and lower coherence, amplitude deviations no longer effectively reflect phase deviations, and phase information is unstable. In these cases, the proposed method effectively solves the problem of identifying observation points, making it a preferable option. Finally, theoretical analysis and experimental results validate the effectiveness of the proposed method for selecting ROPs in low coherence areas.”

Thank you once again for your meticulous review and valuable comments. We believe that these modifications make our research findings more complete and rigorous.

Sincerely,

Author

Reviewer 2 Report

Comments and Suggestions for Authors

The paper has been improved but some conceptual issues have to be clarified.

I completely disagree with your Response 6. If you are working with wrapped phases, any operation of addition or subtraction has to be done in the complex domain. The results can never be between -2pi to 2pi. Imagine that one image has a phase of 179º and the other -179º, the wrapped phases are separated by just 2º but with your operation 358º. The phase probability density functions can be modeled as a Gaussian for low noise levels, as you increase the noise it tends to be a uniform one. This is why for fully decorrelated phases its standard deviation is fixed to 104º.

I suggest you have a look at the book “Radar Interferometry” by Ramon Hanssen sections 4.2, and 4.3. There is a complete and didactic explanation of the stochastic model for radar interferometry.

Only if you work with unwrapped phases, you can add and subtract the phases directly.

Author Response

Dear Reviewer,

First and foremost, I wish to express my deepest gratitude on behalf of our team for the time and effort you have dedicated to reviewing our manuscript. Your positive evaluation and recognition have been greatly encouraging, and your insightful suggestions and valuable feedback have been instrumental in enhancing our manuscript.

In response to your valuable feedback, we have meticulously revised the manuscript. Specifically, we conducted phase difference calculations in the complex domain, with the difference range being from -pi to pi, as per your suggestion. We also carried out experimental verifications using the information you provided, which confirmed that the phase information indeed follows a uniform distribution when the standard deviation is 1.8 and the observation points are close to decoherence. We performed data preprocessing for observation points with a standard deviation below 1.5, , where the phase distribution approaches a Gaussian distribution. Following your comments, we have revised the description to enhance its accuracy. Additionally, we have updated the experimental results and their descriptions, which further improves the accuracy of our method. The revisions in the manuscript are highlighted in blue.

Thank you once again for your professional review and valuable suggestions. We look forward to your feedback on this revision and hope to meet your expectations.

Sincerely,

Author